# Wearable Near-Field Communication Sensors for Healthcare: Materials, Fabrication and Application

**DOI:** 10.3390/mi13050784

**Published:** 2022-05-17

**Authors:** Xidi Sun, Chengyan Zhao, Hao Li, Huiwen Yu, Jing Zhang, Hao Qiu, Junge Liang, Jing Wu, Mengrui Su, Yi Shi, Lijia Pan

**Affiliations:** 1Collaborative Innovation Center of Advanced Microstructures, School of Electronic Science and Engineering, Nanjing University, Nanjing 210093, China; xidisun@smail.nju.edu.cn (X.S.); lhao@smail.nju.edu.cn (H.L.); yuhw@smail.nju.edu.cn (H.Y.); zjing@smail.nju.edu.cn (J.Z.); haoqiu@nju.edu.cn (H.Q.); jgliang@jiangnan.edu.cn (J.L.); wjing@smail.nju.edu.cn (J.W.); sumengrui@outlook.com (M.S.); 2Division of Advanced Nanomaterials, Suzhou Institute of Nano-Tech and Nano-Bionics, Chinese Academy of Sciences (CAS), Suzhou 215123, China; cyzhao2022@sinano.ac.cn; 3Engineering Research Center of IoT Technology Applications (Ministry of Education), Department of Electronic Engineering, Jiangnan University, Wuxi 214122, China

**Keywords:** near-field communications (NFC), energy harvesting, healthcare, wearable materials, wearable electronics

## Abstract

The wearable device industry is on the rise, with technology applications ranging from wireless communication technologies to the Internet of Things. However, most of the wearable sensors currently on the market are expensive, rigid and bulky, leading to poor data accuracy and uncomfortable wearing experiences. Near-field communication sensors are low-cost, easy-to-manufacture wireless communication technologies that are widely used in many fields, especially in the field of wearable electronic devices. The integration of wireless communication devices and sensors exhibits tremendous potential for these wearable applications by endowing sensors with new features of wireless signal transferring and conferring radio frequency identification or near-field communication devices with a sensing function. Likewise, the development of new materials and intensive research promotes the next generation of ultra-light and soft wearable devices for healthcare. This review begins with an introduction to the different components of near-field communication, with particular emphasis on the antenna design part of near-field communication. We summarize recent advances in different wearable areas of near-field communication sensors, including structural design, material selection, and the state of the art of scenario-based development. The challenges and opportunities relating to wearable near-field communication sensors for healthcare are also discussed.

## 1. Introduction

The Internet of Things (IoT) is an information carrier based on the Internet, traditional telecommunications networks, etc. It allows all common physical objects that can be independently addressed to form an interconnected network. The information interaction between people and things and things and things is the core of the IoT. People use sensitive devices such as radio frequency identification (RFID) and intelligent sensors to sense changes in the property states of objects, and transmit real-time information to the receiver to achieve intelligent detection and control. RFID is a significant and widespread application scheme for IoT systems. RFID is not only a communication that can realize two-way communication in the state of contact between two objects, but also has a unique identifier and can be equipped with various sensing functions. Due to these characteristics, RFID plays a key role in people’s daily life and industry, such as cargo tracking, food safety, environmental sensing, security and many other fields [1,2,3]. Since near-field communication (NFC) is developed based on RFID technology combined with wireless interconnection technology, NFC is also considered as a subset of RFID. RFID includes different working frequency bands, such as low frequency, high frequency and ultra-high frequency. NFC uses the 13.56 MHz frequency band for fast communication of short-distance devices [4]. NFC technology is different from RFID in that it can realize two-way interaction between electronic devices, with extremely high security and confidentiality. In wearable device applications, NFC is a promising tool for physiological signal measurement with a battery-free passive mode, wireless communication, fast and contactless data transfer [5,6].

RFID technology was originally developed to replace barcodes and for anti-counterfeiting and anti-theft, and over the last decade or so the technology has been widely used to identify targets and track objects in areas such as automatic parking lot fees, the tracking of goods inventories, and automatic door opening and closing, etc. [7,8,9,10]. Since RFID technology has begun to be combined with sensors [11], the application of RFID sensors has expanded rapidly. The result data as of December 2021 for the Google Scholar search keyword “RFID sensors” are reported in Figure 1a. It can be seen from the figure that RFID sensors have developed rapidly since 2000. The number of searches has fluctuated in recent years. In Figure 1b, as of December 2021, the result data corresponding to searches on Google Scholar for the keyword “NFC sensors” are reported. The number of keyword search entries on Google Scholar for NFC sensors, a relatively new field, is much smaller than the number of entries for RFID sensors. The number of searches on Google Scholar containing the NFC sensors keyword has grown steadily since 2000, with a jump in 2021. From the above, it can be concluded that the rapid development of NFC technology has aroused great interest and the extensive attention of people, and has shown great potential in various wearable devices.

With the continuous development of IoT, wearable electronic devices have experienced a new upsurge in research interest [1]. Wearable electronic devices are electronic devices or medical health devices with embedded sensors, wireless communication and flexible substrates that people can wear directly on their bodies. Wearable electronic devices can sense, record, analyze and maintain health status and intervene, regulate or treat disease. Combining wearable devices and wireless communication data can provide new functions such as real-time analysis, real-time monitoring, timely feedback and dynamic response [12,13,14,15,16,17]. With the rapid popularization of smart phones, smart bracelets and other devices, healthcare is increasingly linked to smart detection wearable devices. At present, wearable NFC sensors have achieved remarkable development in the fields of health analysis, exercise tracking, real-time monitoring, etc. [1,18,19]. NFC sensors are a promising tool for real-time health monitoring, featuring non-contact data transmission under passive tags, wireless communication and a rapid response [20,21,22]. Wearable NFC sensors are now widely used in people’s daily lives, and there are great prospects for commercial applications in smartphone sensing, food safety and health monitoring [23].

In this review, we will comprehensively review the research results of wearable NFC sensors and their application in healthcare (Figure 2). First, the design and preparation methods of wearable NFC antennas will be systematically summarized. Then, the sensing types and data transmission methods of wearable sensors are introduced. Finally, we will present the applications and prospects of wearable NFC sensors for healthcare. The future development of wearable NFC sensors is discussed.

## 2. Wearable Materials

The various wearable sensors are currently complex, and have been developed to be portable and miniaturized. However, the cornerstone materials are mechanically rigid and based on flat designs. This results in a large number of wearable electronic devices that cannot be fully integrated with the soft human body. New wearable devices need to be soft, beyond traditional rigid wafer and flat circuit board technology. Such wearable devices need to be soft, thin, inexpensive, and durable, while maintaining excellent sensing performance.

Parylene-C has been widely used as a surface coating for various materials due to its excellent physical and chemical properties, such as superior flexibility, high transparency, easy preparation, and good chemical stability in organic solvents. At the same time, parylene-C films can be used as new wearable materials due to their ultra-thin, highly flexible and easy-to-peel properties. Traditional methods of fabricating flexible devices require the introduction of a sacrificial layer between a flexible substrate and a rigid support. However, this method is inefficient and usually takes tens of minutes to hours because of the limitations of slow reaction and diffusion. In addition, the dissolution of the sacrificial layer usually requires harsh etchants (e.g., HF, NaOH, etc.) that may adversely affect electronic devices and systems. Parylene-C film can be peeled off from a rigid substrate by means of a capillary-assisted electrochemical delamination approach without damaging the performance of the electronics on the substrate (Figure 3a). An ultrathin polymer film is prepared on a heavily doped Si wafer as a rigid support. Thereafter, positive and negative potentials are applied to the Si wafer and NaCl electrolyte aqueous solution, respectively. The positively polarized Si wafer induces an anodic reaction: Si − 8e^−^ + 8OH^−^ → H_2_SiO_3_ + 3H_2_O + 2O_2_. As the reaction proceeds, the parylene-C film completes the separation from the supporting Si wafer without any significant mechanical damage [24]. Wu et al. fabricated an ultrathin (2.4 μm) perovskite array photodetector (10 × 10 pixels) using parylene-C as the substrate and encapsulation layer [25]. The device demonstrates robust mechanical stability under bending or 50% compressive strain. The parylene-C film can be scooped and laminated on various irregular 3D surfaces covered by small features, forming an intimate contact. Moreover, it can be bent and compressed, akin to an elastomer, without peeling off the skin (Figure 3b).

A conductive polymer is an additional type of wearable material, which has been widely used in wearable sensors. Conductive polymers are different from conventional organic polymers, as they have high electrical conductivity, electron affinity and redox activity. The structures of several conventional conducting polymers, such as poly(3,4-ethylenedioxythiophene) (PEDOT), poly(p-phenylene vinylene) (PPV), poly(thiophene) (PTh), and poly(aniline) (PANI), are given in Figure 4. Both the monomer of the conductive polymer and the polymer itself can be functionalized with different groups. The addition of substitution groups not only simplifies processing and adds functionality, but also improves the electronic properties of the main polymer chain and enhances electrical stability. For example, PEDOT-PSS is essentially a macromolecular polymer consisting of positively charged PEDOT and negatively charged poly(4-styrenesulfonate) (PPS) [26]. Unfortunately, these conducting polymers are not soft and are difficult to stretch while maintaining electrical conductivity due to the conjugated aromatic ring structure. To achieve high electrical conductivity, the chains of the conducting polymers should be arranged with high crystallinity, which causes them to lose their flexibility. Therefore, improving the stretchability of conductive polymers is a major challenge.

A hydrogel is another type of wearable material. Similar to ionic liquids, hydrogels are typically conducted through ionic motion in a hydrated polymer matrix. The modulus of hydrogels may closely matches human skin, allowing for the establishment of conformal contact. However, the conductivity of conventional hydrogels is very low due to the lack of electron conductivity. Conductive polymer hydrogels are an emerging type of hydrogels which combine conductive polymers and hydrogels, offering broad tunability in terms of mechanical properties, conductivity and functionality [27,28,29]. For example, conductive polymer hydrogels can also be obtained by mixing the additive dimethyl sulfoxide (DMSO) into the PEDOT: PSS dispersion followed by controlled dry annealing and rehydration [30]. The strong polar solvent DMSO increases the electrical conductivity of PEDOT: PSS by doping and can extend the PEDOT: PSS microgel particles from the folded state to linear long chains. Meanwhile, the enhancement of the π-stacking crystallinity of PEDOT chains during dry annealing facilitates the crystallization into nanoprotofibers. This interconnected nanoprotofibril morphology can potentially provide better electrical conductivity and mechanical properties by forming a more efficient pathway for electron transfer and sustaining mechanical forces (Figure 5).

## 3. Near-Field Communication

Combining wearable devices with wireless communication data transmission provides new capabilities for real-time monitoring and dynamic responses. Wireless communication technologies, such as RFID, NFC and Bluetooth, have been widely used for information transmission in wearable electronic devices. Among them, NFC is a promising tool for health monitoring. It has features such as being battery-free in passive mode, providing wireless communication, being fast and providing contactless data transmission. Peer-to-peer (P2P) communication is possible between NFC tags and smartphones, which is not possible with other RFID technologies. Additionally, unlike Bluetooth, NFC tags can be used without batteries, which promotes the development of wearable devices in healthcare.

### 3.1. Wearable Communications and Data Transmission

Real-time applications of monitoring are significantly dependent on the NFC or RFID used to monitor and transfer the recorded data. Table 1 shows several wireless communication technologies. As part of wearable sensors, wireless communication technology is required to seamlessly stream important information to the users. RFID technology has been applied in the market and several commercial sensors have been developed in inventory and supply chain applications [17,18,19]. Ultra-high-frequency RFID (UHF RFID) has a long transmission range, but the readers are very expensive (USD 1000–2000). UHF RFID is also more susceptible to environmental impacts, leading to losses and detuning. Low-frequency RFID (LF RFID) has been commercially available for a long time and has a wide range of applications in the market. LF RFID has excellent anti-interference performance, but it has the disadvantages of relatively low transmission speed and a short working range (50 cm). The wireless communication of Bluetooth (3–200 m) has a greater read range and requires less power than NFC. However, Bluetooth technology requires an external battery to power it. NFC are the better protocols for use in transmitting data from wearable sensors due to their low cost, low power consumption and portability, which allows developers to achieve flexible and comfortable wearable sensors.

The ultimate goal of wireless communication technology is to transmit relevant information data to the user or clinician. Therefore, the type of wireless communication should be suitable for the working requirements of wearable sensors. For example, sensors for the detection and care of newborns, especially the health detection of premature babies, require low power consumption and miniaturization [20]. NFC chips with low power consumption and energy harvesting capability may be a promising option.

### 3.2. Energy Harvesting

The issue of power consumption has always played an important role in the development of wearable devices. Batteries in many wearable devices should be managed as hazardous waste because they are toxic or reactive. Therefore, low-power and green energy wearable electronics have attracted widespread attention and fit the concept of green social development. In semi-passive mode, the NFC sensor can operate independently in autonomous and continuous monitoring. In battery-free (passive) mode, the tag is completely passive. In this mode, the NFC sensor collects energy from the incoming RF emissions to power the sensor interface and RF transmissions. The lifetime of the wearable sensor tag may include operation in two modes: in semi-passive mode until the battery is exhausted, and thereafter in passive mode. Data are stored in non-volatile memory and retained while the device is not powered on. NFC sensors with energy harvesting capabilities can be obtained from many sources (e.g., vibration, light, sound, heat, and temperature changes). In passive or semi-passive mode, NFC tags can harvest energy from the magnetic field generated by the reader to power external devices.

### 3.3. NFC Antenna Design

Antenna is an important component of NFC sensors, and its main function is to transmit radio frequency signals between the tag and the reader. The conversion efficiency of its electromagnetic energy plays a significant role in the applicability of NFC sensors [21]. The wearable NFC tag must not affect the comfort level of the wearer under any circumstances. This not only requires a flexible, small-size, and lightweight design, enabling rational integration, but also demands sustainable operation to avoid the need for frequent battery recharging. In addition, a stable and controllable wireless communication link is required to provide the wearer with the freedom of movement, without being out of read range. To meet these different requirements, the design of the antenna is particularly important. A mobile user prefers a small-sized antenna for wearable integration, whereas an antenna designer will opt for larger dimensions of about half a wavelength to obtain better overall performance. A reasonable design must reconcile these conflicting demands to achieve optimal device performance with manageable dimensions [22].

The design (size and shape) of NFC tags antenna is a significant aspect for wearable NFC sensors [23]. The size of the coil is a decisive factor in the preparation of wearable NFC sensors due to the limitations of operating frequency and communication distance. A suitable size of loop coil determines its quality factor (Q factor) and read range. Hirayama et al. reported a study on the coupling between two toroidal loop antennas to investigate the importance of antenna size [30]. The coupling factor K is a function of radius r_1_ and r_2_ between the two R for different axis distances X of the two loop coils. The optimal case is when the two loop coils have approximately the same radius (r_1_ ≈ r_2_). Additionally, detuning of the NFC antenna, reducing wireless power transfer, will usually occur when the mobile phone is close to the NFC tag. The Q factor of antenna can be obtained from Q = Im (Z)/Re (Z). With the increase in distance, the Q factor and input voltage of the NFC antenna decrease and the bandwidth increases. The resonant frequency decreases due to the change in chip capacitance. Therefore, the maximum distance between the tag and the reader to acquire the tag is 2 cm in the best-case scenario for both analyses.

### 3.4. NFC Antenna Manufacturing Techniques

Manufacturing on flexible substrates remains the major challenge in fabricating wearable NFC and RFID sensors and electronics. Currently, lithographic processes are the mainstream fabrication method. Lithography is the most stable and precise method to fabricate electronics with high controllability (Figure 6a) [31,32,33,34]. However, lithography techniques have a high cost because they need expensive photo-resistant materials, complex processes, expensive equipment, and strict working environments.

Compared with lithography, printing technologies are more effective and popular due to their simplified processing steps, low cost, and simple patterning techniques. Screen printing is the most attractive approach to produce a large area of electronics. Ink is transferred to substrates such as paper and ceramics through the aperture of a stencil by pressure-forming various images, when the printing plate is printed (Figure 6b) [35]. A copper ink has been used to fabricate NFC antennas by means of screen printing [36]. The inductance of the copper ink antenna was similar to that of the silver paste antenna with a return loss of −16.8 dB, which is lower than −8.8 dB of the silver-tin antenna. The copper-ink antenna has a wider available range of frequency bands compared to the copper-etched antenna. However, the Q factor of the copper ink antenna is too low to be used in conventional copper-etched antenna (~7.8) because of its low thickness. To solve the disadvantage of a low Q factor, multiple printing of copper ink is required, or the copper ink should be modified to enable thicker printing. In many wearable electronics, antennas have to withstand harsh environments; therefore, metallic nanoparticles are also used in inks used for printing antennas.

Inkjet printing is currently one of the most promising techniques that could revolutionize large area electronics fabrication, and its advantage for the fabrication of flexible antennas is that the thickness of the printed coil can be controlled. Ortego et al. introduced an analysis of an inkjet-printed coil antenna (Figure 6c) [37]. When the inkjet-printed antenna is finished, the thickness of the antenna can be measured by a P-16 + Profiler. The Q factor and impedance of antennas can be calculated by the different thickness of the antenna. If the thickness of the antenna is under 40 μm, the Q value is up to 90 in theory. Wang et al. introduced a reliable and low-cost solution treatment process for preparing flexible metal antennas with high adhesion and low resistivity on RFID tags by inkjet printing combined with surface modification and chemical deposition (Figure 6d) [38]. The thickness of the copper layer can be adjusted by inkjet printing; the reflection decreases by more than 15 dB when the copper layer thickness is lower than 1.1 μm. However, the return loss of the antenna decreases when the thickness of the copper layer is higher than 1.1 μm. Therefore, the antenna shows the best performance when the copper layer thickness is about 1.1 μm. Thus, inkjet printing technology is proved to be a most promising technology for the fabrication of NFC antennas, and due to its convenient manufacturing process and flexible substrate, it is a perfect candidate for wearable NFC terminals. However, how to improve the conductivity of inkjet-printed patterns of antennas is still a pressing challenge that needs to be solved.

## 4. Wearable NFC Sensors for Healthcare

Flexible electronics as an emerging technology trend is important for economic growth and improving the quality of human life. As the demand for portable, lightweight, and low-cost flexible and wearable electronics continues to grow, significant research attention is needed to address future challenges in the construction of next-generation electronic devices to achieve the necessary technological advances in performance characteristics and a wide range of potential applications. With the continuous innovation of materials and the continuous optimization of NFC/RFID structures, more and more novel designs and various applications of wearable NFC/RFID sensor technology have been reported [38,39,40,41,42,43,44,45]. The emergence of IoT has strengthened the connection between things and people and things. Combining wearable electronics and NFC/RFID can provide more convenient and novel functions for sensors. Compared with RFID sensors, NFC sensors have many advantages, such as fast transmission speed, low power consumption, high security and confidentiality. At present, NFC sensors are integrated in most smart phones and smart bracelets, so they have great development prospects and wide development space in the wearable field. Integrating NFC technology with sensing units that have the ability to detect physical and chemical signals in a single wearable sensor allows for different kinds of health detection [43,46,47,48,49,50,51,52].

### 4.1. Types of Sensing Using Wearable Sensors

The sensing element is the core component of an NFC sensor. With the appropriate choices of flexible materials and design, wearable NFC sensors may be fabricated by using different signal transduction methodologies, such as capacitance, resistive, piezoelectric, and triboelectric methods [53]. Recently, techniques different from the traditional electrical signal-sensing methods have been reported—these transductions may be achieved with ions [54]. The electrical double layer behaves similarly to a supercapacitor, and supercapacitor sensors based on this may offer a three-orders-of-magnitude higher response than traditional capacitance sensors [55]. When ionic gels are used as electrolytes for supercapacitor sensors, they are more beneficial for wearable sensors.

Based on the type of sensing signal, wearable NFC sensors for healthcare may be divided into two kinds: biophysical signals sensors and biochemical signals sensors. It seems that this classification is better for us to design wearable NFC sensors for healthcare. Biophysical signal sensors are the most standardized of the applications for wearable sensors [56,57,58,59,60,61,62,63,64,65,66,67]. Body temperature, blood pressure, pulse, and muscle stretching, etc., are examples of physical signals directly related to people health. Although it is easy to monitor such vital signs in the hospital, it is challenging to measure them anytime anywhere. Currently, many wearable devices can be used to address this challenge to some extent, but poor wearability and signal transmission prevent these products from reaching medical standards. It is exciting to note that there is significant academic interest in developing wearable pressure, strain, and temperature sensors. A wide range of new materials such as silver nanowires, carbon nanotubes, graphene, liquid metals, and conducting polymers have been used to design wearable physical sensors [68,69,70,71]. Current sensors recognize local physical stimuli throughout the body. High-resolution sensors with millions of pixels to map such biometric signals remain a major challenge. Electrophysiological signals (EPS) are special kinds of biophysical signals. EPS sensors measure the electrical potential difference between electrodes in specific tissues such as the brain, muscle, and heart. These measurements are called electroencephalographs (EEG), electromyograms (EMG), and electrocardiographs (ECG), respectively. The core issue for all EPS is to design biocompatible, thin, and flexible epidermal electrodes to decrease skin–electrode interface impedance. Thin film electrodes are the most common flexible sensor electrodes; however, due to the roughness of the skin surface, it is difficult to make complete contact with the skin. The presence of an air gap increases the interfacial impedance, which leads to significant ambient noise (Figure 7a). Conductive hydrogel electrodes create a tight conformal contact with rough skin surfaces, but water loss in the gel may prevent long-term use of the electrodes, and ions in the gel may damage the skin (Figure 7b). Microneedle electrodes are also a good solution (Figure 7c). The microneedles can penetrate into the stratum corneum, but do not come into contact with the nerves. Therefore, this type of contact will not make the wearer feel uncomfortable. The length of the microneedles needs to be matched to the thickness of the stratum corneum, enabling their direct contact with the epidermis, thereby reducing interface resistance.

Electrochemical signal sensors are one of the most common types of wearable sensor over the years. Chemical sensors may account for at least 40% of the wearable market share to date [72]. The wearable NFC sensors, with their exclusive data transmission, chemical and electronic properties have been the best choice to carry out various types of biochemical sensing. The common types of biochemical sensors include monitoring of the pH of sweat, glucose, cholesterol, lactic acid, etc. [73,74,75,76,77,78,79,80,81,82,83,84,85,86]. In a single graphene sheet, three out of the four outer electrons of one carbon atom form three σ-bonds, while the remaining electron forms a vertical π-bond. When curving a graphene sheet into a carbon nanotube (CNT), the σ-bonds in the CNT shell slightly deviate from the planar surface and the π-bonds orient to the outside (Figure 8) [86,87]. Since the curvilinear sidewalls and hydrophobicity of CNTs provide strong interaction through the π-bond, they can be used to develop glucose and pH sensors [88]. Cholesterol is a lipid formed on the membranes of human cells. Sensors for monitoring cholesterol are manufactured by integrating single-walled carbon nanotubes and multi-walled carbon nanotubes with sol-gels. These sensors can be developed with techniques such as screen-printing, where a separate membrane of cholesterol esterase and cholesterol oxidase is attached to the sensing surface [89]. Using conducting polymers as sensing films, DNA, proteins, and glucose can be detected via a label-free electrochemical read-out. Conductive polymers use the interaction between their surface and biomolecules for physical adsorption. Since conductive polymers can carry a large charge, electrostatic interactions between cationic conductive polymers and anionic biomolecules play a central role. However, physical adsorption also includes other interactions, especially in the adsorption of proteins or antibodies, for example, hydrophobic forces and van der Waals forces.

In practical applications, it is very promising if multiple sensing modalities can be integrated into a single sensing platform [90,91]. It is worth noting that not all sensing elements can be integrated with NFC. The readers must be able to distinguish between acceptance information and identity information. Because of the limitations of NFC operating frequency and communication range, there are currently fewer types of NFC-based sensors, so it is necessary to explore and develop more types of wearable NFC sensors.

### 4.2. Weak Electrical Signal Processing

For wearable devices, weak electrical signal processing and displays are very important. Many wearable devices require low-voltage operation while providing high enough current to drive the circuits and high gain to amplify small signals. Wearable electronic devices can accomplish real-time sensing on human tissues, presenting better signal quality and comfort. Within this scope, flexible amplifiers are promising candidates for signal recording, stabilization, and noninvasive in close proximity to the site of interest [92,93,94,95,96,97]. Requirements for such electronic devices include a wearable substrate, low-voltage operation, and a small size for high-resolution signal mapping for a reliable representation of information. A conventional signal acquisition process involves the weak electrical signal being amplified, converted to a digital signal, and then processed into a digital output (Figure 9a) [93]. Wearable sensors require weak signal amplifiers with both high current output and high signal gain. Wang et al. combined thin film transistors with ferroelectric HZO gating to demonstrate wearable sub-thermionic organic transistors as well as ultra-high-gain amplifier circuits [95]. On an ultra-thin polyimide substrate, the film was highly transparent and flexible to the epidermis to allow the construction of skin-like wearable electronic devices. The device contains an enhancement–depletion mode inverter using two sub-thermionic organic thin film transistors with different dimensions (Figure 9b). This inverter exhibited full swing output near zero input voltage with a peak power of ~50 nW. Remarkably, this device obtained giant voltage gain (A_v_) of 4.1 × 10^3^ (1.1 × 10^4^) under V_dd_ = −1 V(−3 V).

### 4.3. Wearable NFC Sensors for Biophysical Signals Monitoring

Information on human health status can be obtained directly from physical signals generated by the body (e.g., pulse, heart rate and body temperature). Thus, biophysical signal monitoring is the mainstream research direction of wearable NFC sensors [98,99,100,101]. Continuous monitoring of vital signs is essential in neonatal care, especially in cases of severe prematurity. Current health monitoring platforms require multiple sensors hard-wired to a neonate’s fragile skin and invasive lines inserted into their tender blood vessels. In response to this challenge, John et al. developed an NFC sensor for neonatal detection and care, especially the health detection of premature infants, which can monitor the vital signs of neonates in real time by only requiring water to adhere to the skin (Figure 10a). The device consists of a set of 50 mm–100 mm-wide, millimeter-thick serpentine copper wires that interconnect multiple chip-scale components. The device also contains a loop antenna to implement the NFC sensor, and is configured for real-time wireless data transmission and wireless power supply (Figure 10b). Computational facilities on the NFC system-on-a-chip of the ECG electronic system can support the Pan–Tompkins algorithm for accurate, analysis of electrocardiogram signals in real time to yield heart rate and heart rate variability on a beat-to-beat basis (Figure 10c) [98]. Bao et al. reported a body area sensor network composed of chip-free and battery-free stretchable sensor tags. The wearable sensor consists of a flexible inductor, a resistive strain sensor, and a stretchable capacitor. Since there are no chips and batteries inside the on-skin flexible sensor tags, this improves the conformability and draftability of the design. The body area sensor network concept was realized with multiple sets of flexible sensors and soft readout circuits located on clothes. One node was chosen to be located at the wrist for pulse detection, another node at the abdomen for breath detection and the other three nodes at the elbow, left leg, and right leg for body movement detection. The detected signal can be transmitted to a phone via RFID. The sensor network can continuously detect critical biophysical signals (respiration, pulse, etc.) and therefore has the potential to be used for real-time monitoring and healthcare in a novel personal health monitoring system [102].

Body temperature is also an essential signal for human health detection. Temperature sensors placed at different locations on the body can measure various parameters such as thermoregulation, heat expenditure, and wound healing [103,104,105,106]. Krishnan et al. developed a wearable, epidermal wireless thermal sensor. The NFC sensor consists of an electronic device for NFC-based data transmission and analog signal conditioning and a Cr/Au-based flexible temperature sensor. Since the sensor is affected by the temperature coefficient of the resistance of the metal film, the change in temperature causes a change in the resistance (Figure 10d). This sensor can be operated in battery-free mode through smartphones, where data and power transmission are achieved by means of resonant inductor coupling using NFC protocols (Figure 10e) [107]. In addition, when the sensor collects information, NFC will transfer the collected data to the smartphone to achieve real-time monitoring and analysis of people’s health.

### 4.4. Wearable NFC Sensors for Biochemical Signals Monitoring

Chemical signals from the human body are also an important part of the detection of wearable sensors. Wearable electronic devices can collect vital information from the sweat, saliva, tears, and blood to predict the occurrence and development of diseases [108,109,110,111,112,113,114]. Wearable NFC sensors based on biochemical signal monitoring can collect sensitive substances from secretions on the body surface, predict the occurrence of diseases and analyze health status. NFC sensors for the direct detection of biochemical signals have great application prospects, but the types and numbers of such NFC sensors are still relatively small. Sweat is the most common substance in human secretions; therefore, sweat sensors have become one of the most significative developing directions of wearable diagnostic devices. Bandodkar et al. reported a wearable NFC sweat sensor that can monitor sweating rate/loss, pH, lactic acid, glucose and chloride. The device includes two components: a flexible, microfluidic network and a light, wearable NFC electronic module. The disposable microfluidic network houses the various chemical sensors, and it can support switches, channels, and reservoirs to handle small amounts of sweat delivered to the system by the action of the gland (Figure 11a). For reuse, the NFC electronic module is mounted on a disposable microfluidic system with a releasable electromechanical interface (Figure 11b). Rigorous two-day field studies and the correlation of data acquired by glucose sweat sensors with blood levels demonstrated both the potential for real-time use and the potential for the long-term tracking of blood analyte concentrations (Figure 11c) [115].

Madhvapathy et al. reported a soft, battery-free, noninvasive, reusable skin hydration sensor adherable to most of the body surface (Figure 12a). The entire device contains electronics for sensing and wireless communication. The sensing module comprises two commercial thin-film resistors connected in series to form a heater and a surface-mounted negative temperature coefficient thermistor (Figure 12b). The antenna tuned to 13.56 MHz (NFC frequency) connects to the RF microcontroller to power the sensor and to communicate data to the smartphone using NFC protocols (Figure 12c) [116]. This work has the potential to create opportunities for patient measurement in clinical settings and in the home. Human tears contain numerous chemicals such as water, glucose, and lactic acid, and measuring the concentration and composition of these substances can often help to assess health conditions and predict disease, including diabetes and glaucoma. Ali et al. reported a wearable NFC tear sensor that can analyze the electrolyte composition of tears. The microfluidic device was placed in a portable readout device consisting of various LED illumination wavelengths for fluorescence excitation. A smartphone app was used to capture fluorescence detection images, which were digitally processed to obtain concentration values for quantitative analysis of electrolytes in artificial tears [117]. However, the miniaturization and multifunctionality of the wearable NFC sensor is still a major challenge.

Although wearable NFC sensing technologies are diversely reported, more clinical tests are needed for their application in health detection, medical treatment, and disease prediction to improve the accuracy and reliability of these sensors.

## 5. Summary and Perspectives

Although NFC technology was introduced in 2004, it was only designed to be applied to healthcare in recent years. UHF RFID technologies have long been used on the market, and some commercial sensors have been developed in inventory, logistics, and supply chain applications. However, UHF RFID readers are much more expensive, leading to difficulties in the profits of commercial products. On the other hand, UHF RFID is more susceptible to environmental influences that can lead to losses and detuning. The cost of RFID sensors is mainly dependent on the cost of tags (chips). Thus, the cost of chipless RFID devices is much lower than that of traditional RFID devices. These chipless sensors are not only low-cost, but also highly reliable, even if these devices operate in a harsh environment. However, chipless RFID lacks commercial standards and requires a dedicated reader to interrogate the tags. To date, the only commercially available chipless RFID tag is the SAW tag. Bluetooth has a longer reading range (10–100 m) and lower power consumption than NFC. However, Bluetooth technology requires a battery to power it because it has no ability of energy harvesting. The costs of NFC readers are usually lower than other RFID sensors. Moreover, NFC technology can use batteries to transmit energy. In passive mode, NFC tags can harvest energy from the readers to power external electronics. Many sensitive NFC with energy harvesting capabilities are already on the market. NFC technology can facilitate the development of low-cost and green energy wearable devices.

Wearable NFC sensors are gaining more and more attention and being developed in many applications. This is due to their simple manufacturing methods and the fabrication cost of the substrates. With the development of IoT and smartphones, wearable electronic devices have broad potential and applications in the fields of healthcare, real-time monitoring, food safety, electronic skin, and human–computer interaction [118,119,120,121,122,123,124]. Combining NFC technology and wearable electronic devices has become a popular research direction at present. Here, we provide a comprehensive overview of materials and technologies required for the next generation of wearable wireless sensing technology. Flexible and small antennas have been successfully used for wireless signal readout and data processing. Significant progress has been made in the manufacture of wearable sensors using a variety of flexible materials, including carbon materials, conductive polymers, ionic gels and liquid metal. Successful attempts have also been made to integrate new flexible materials into hybrid electronic systems, enabling the collection of various data wirelessly.

Nevertheless, as wearable electronics require continuous testing, energy devices are an urgent problem to be solved. Micro-battery delivery systems, flexible large-capacity batteries, supercapacitors, and self-powered systems are the current solutions [125,126,127,128]. Additionally, the selection of biodegradable and non-toxic materials for wearable sensors is also an issue to be considered for their commercialization. Finally, the integration of individual NFC sensors into a sensor array facilitates all-round health detection and real-time analysis of the human body, and also opens up new frontiers for wearable electronic devices.

## Figures and Tables

**Figure 1 micromachines-13-00784-f001:**
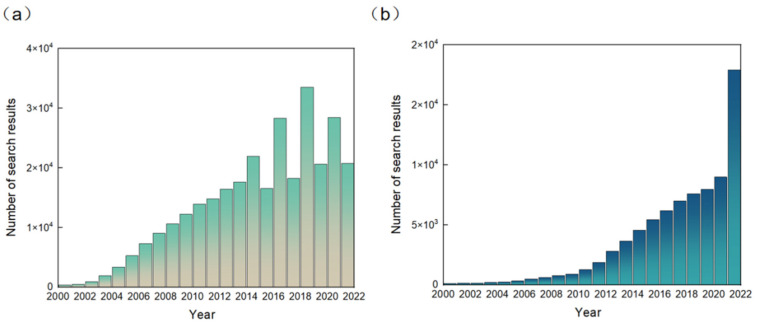
Number of (**a**) search results on Google Scholar in December 2021 according to the keyword “RFID sensors”. Number of (**b**) search results on Google Scholar in December 2021 according to the keyword “NFC sensors”.

**Figure 2 micromachines-13-00784-f002:**
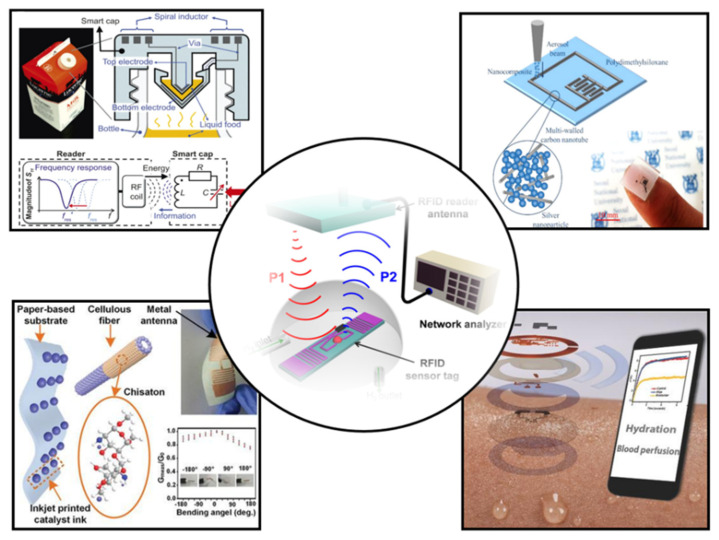
Recent progress in NFC and RFID sensors [12,13,14,15,16]. Reprinted with permission from Ref. [12]. Copyright© 2022, Springer Nature. Reprinted with permission from Ref. [13]. Reprinted with permission from Ref. [14]. Copyright© 2022, American Chemical Society. Reprinted with permission from Ref. [15]. Copyright© 2022, Royal Society of Chemistry. Reprinted with permission from Ref. [16]. Copyright© 2022, American Chemical Society.

**Figure 3 micromachines-13-00784-f003:**
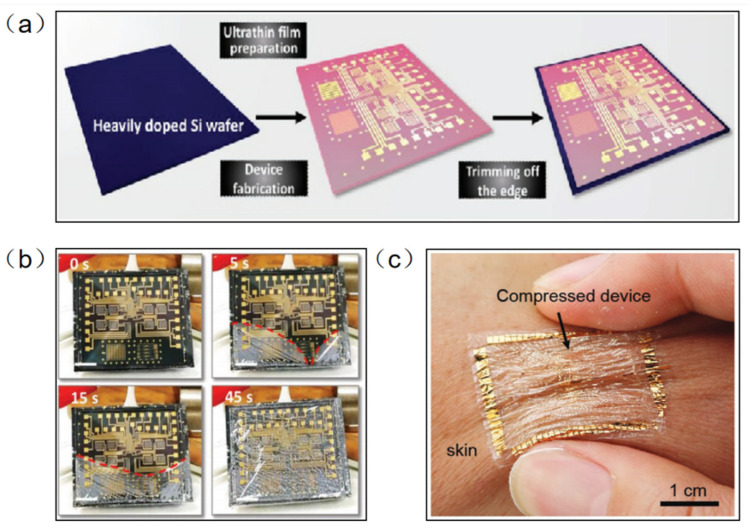
(**a**) Electronic film preparation on a heavily doped Si wafer [24]. (**b**) The progress in the detachment of an ultrathin parylene-C-based electronic foil [24]. (**c**) The ultrathin device is laminated on the skin and survives being compressed, similarly to the elastomers [25]. (**a**,**b**) Reprinted with permission from Ref. [24]. Copyright© 2022, WILEY-VCH Verlag GmbH & Co. KGaA, Weinheim. (**c**) Reprinted with permission from Ref. [25]. Copyright© 2022, WILEY-VCH Verlag GmbH.

**Figure 4 micromachines-13-00784-f004:**
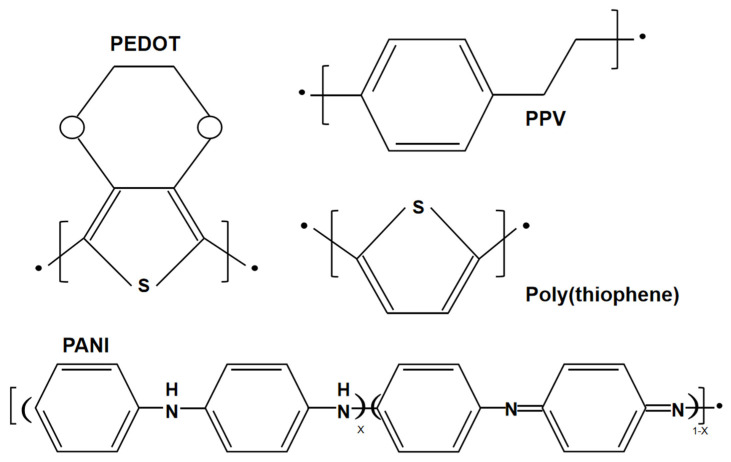
Structures of poly(3,4-ethylenedioxythiophene) (PEDOT), poly(thiophene) (PTh), poly(p-phenylene vinylene) (PPV), and poly(aniline) (PANI).

**Figure 5 micromachines-13-00784-f005:**
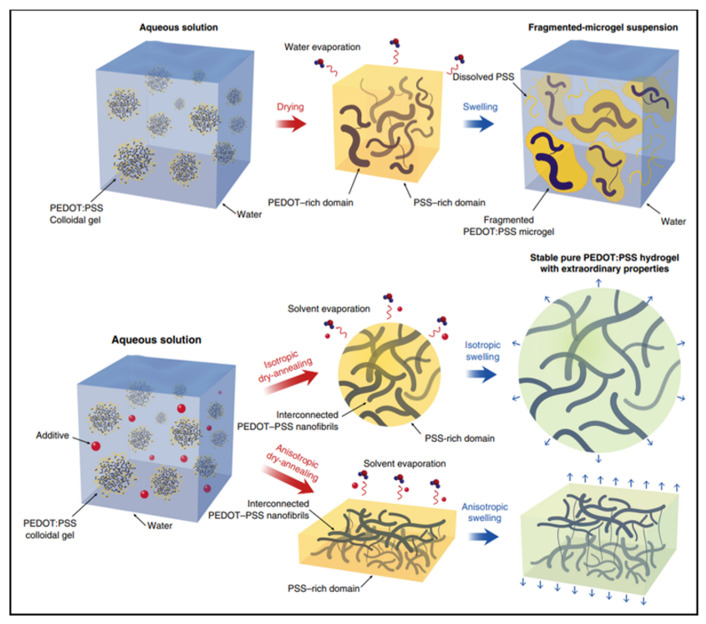
I. Typical drying and swelling processes of pristine PEDOT:PSS without DMSO. II. Dry-annealing and swelling processes of PEDOT:PSS with DMSO as the additive [30]. Reprinted with permission from Ref. [30]. Copyright© 2022, Springer Nature.

**Figure 6 micromachines-13-00784-f006:**
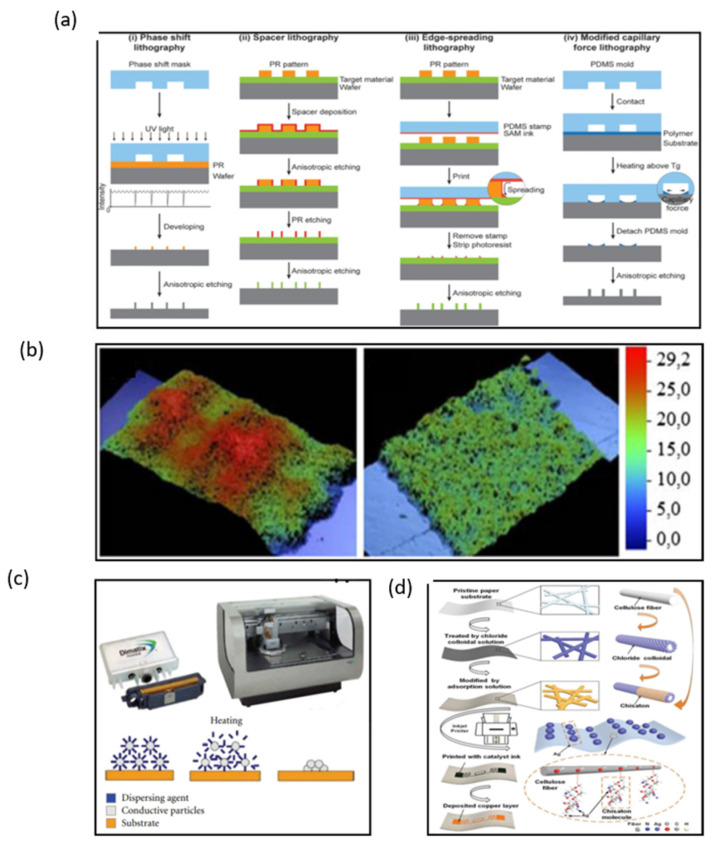
(**a**) Four different edge lithography approaches based on methods for utilizing the edge [32]. (**b**) Microphotographs of screen-printed lines with a mesh of 43 T/cm and 140 T/cm [35]. (**c**) Cartridge and Printer Dimatix 2831 and sintering process [37]. (**d**) Fabricating flexible RFID tags on paper substrates via inkjet printing, surface modification, and electroless deposition [38]. (**a**) Reprinted with permission from Ref. [32]. Copyright© 2022, John Wiley and Sons. (**b**) Reprinted with permission from Ref. [35]. Copyright© 2022, Springer Nature. (**c**) Reprinted with permission from Ref. [37]. Copyright© 2022 I. Ortego et al. (**d**) Reprinted with permission from Ref. [38]. Copyright© 2022 WILEY-VCH Verlag GmbH & Co. KGaA, Weinheim.

**Figure 7 micromachines-13-00784-f007:**
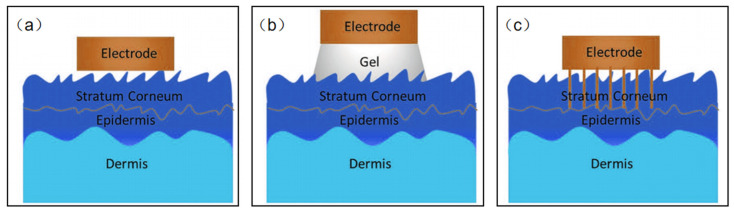
(**a**–**c**) Schematic diagram of the electrode–skin interface model.

**Figure 8 micromachines-13-00784-f008:**
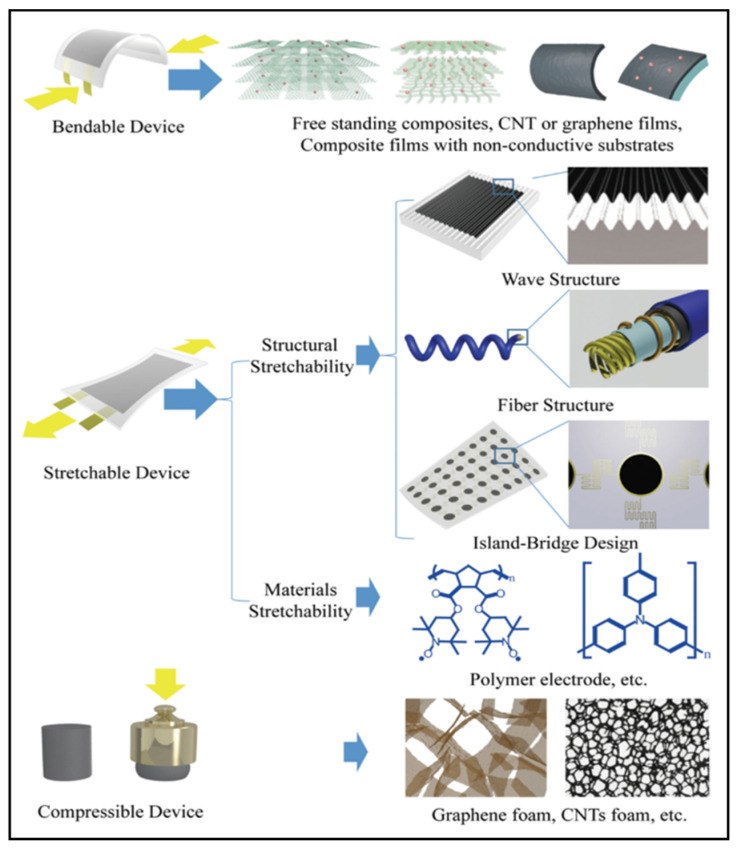
Structures and applications of CNTs and graphene [87]. Reprinted with permission from Ref. [87]. Copyright© 2022 WILEY-VCH Verlag GmbH & Co. KGaA, Weinheim.

**Figure 9 micromachines-13-00784-f009:**
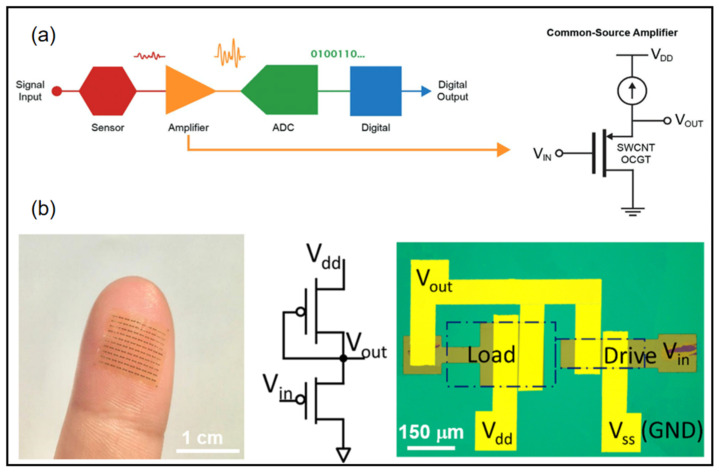
(**a**) Overview of common signal acquisition systems [93] (**b**) Flexible device array laminated on a fingertip (left). Equivalent circuit diagram and optical microscope image of an enhancement–depletion mode amplifier [95]. (**a**) Reprinted with permission from Ref. [93]. Copyright© 2022, WILEY-VCH Verlag GmbH. (**b**) Reprinted with permission from Ref. [95]. Copyright© 2022, Springer Nature.

**Figure 10 micromachines-13-00784-f010:**
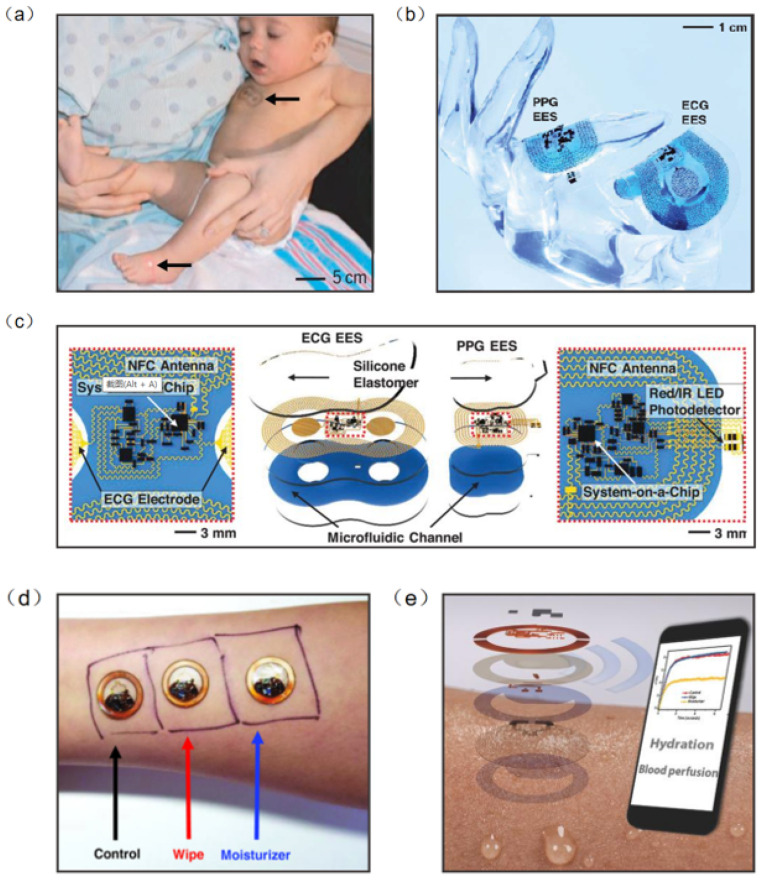
(**a**) A neonate with an ECG device on the chest [98]. (**b**) The device can be overlaid on a life-size finger [98]. (**c**) Schematic illustration of wireless, battery-free modules for recording ECG and PPG data and skin temperature [98]. (**d**) Wireless measurements of skin hydration on human subjects [107]. (**e**) Exploded-view schematic illustration of an epidermal wireless thermal sensor [107]. (**a**–**c**) Reprinted with permission from Ref. [98]. Copyright© 2022, The American Association for the Advancement of Science. (**d**,**e**) Reprinted with permission from Ref. [107]. Copyright© 2022, WILEY-VCH Verlag GmbH & Co. KGaA, Weinheim.

**Figure 11 micromachines-13-00784-f011:**
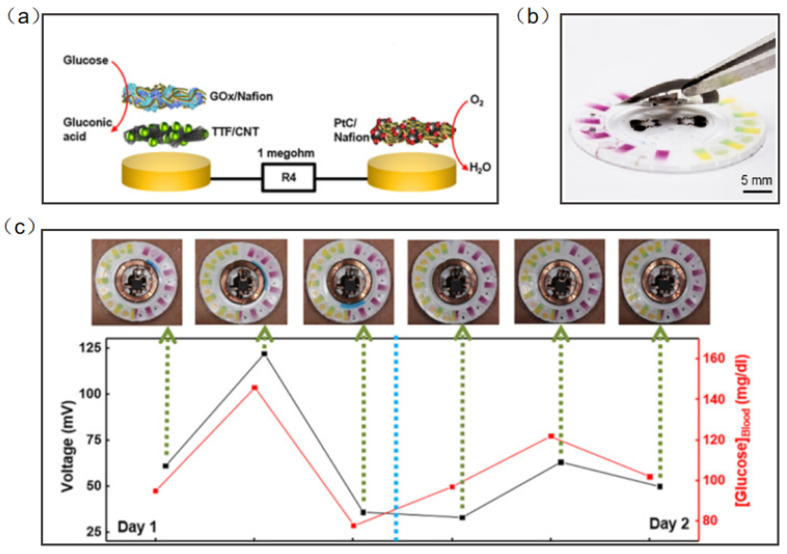
(**a**) Exploded-view schematic illustration of the layer makeup of the biofuel cell-based glucose sensor [115]. (**b**) The exploded view of the complete hybrid battery-free system [115]. (**c**) Correlation of data acquired from glucose sensors with those acquired from blood glucose over a period of 2 days for subject [115]. Reprinted with permission from Ref. [115]. Copyright© 2022, The American Association for the Advancement of Science.

**Figure 12 micromachines-13-00784-f012:**
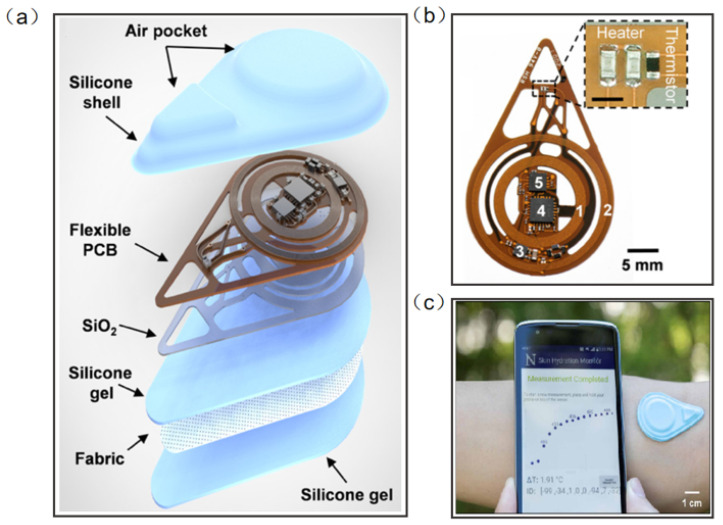
(**a**) Diagram illustrating the layers and components of the device [116]. (**b**) Photograph of a fully assembled flexible printed circuit board for the device [116]. (**c**) The exploded view of the device. Sensors on smartphones provide visualization of actual measurements [116]. Reprinted with permission from Ref. [116]. Copyright© 2022, The American Association for the Advancement of Science.

**Table 1 micromachines-13-00784-t001:** Comparison of several wireless communication technologies.

Type	Band	Applications	Range	Reader Cost
Bluetooth	2.4 Ghz	Wireless Sensors	3–200 m	Low
NFC	10–15 MHz	Wireless Sensors	10 cm–1.5 m	Low
LF RFID	120–500 KHz	Wireless Sensors	50 cm	High
UHF RFID	433–950 MHz	Wireless Sensors	3–100 m	High, $1000–2000

## Data Availability

Not applicable.

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
