# Peer review of "Wearable Near-Field Communication Sensors for Healthcare: Materials, Fabrication and Application"

_micromachines, 2022, doi:10.3390/mi13050784_

Round 1

Reviewer 1 Report

  1. This review focus on NFC-related materials, fabrication, and application in the field of wearable healthcare sensors. The title of the manuscript should be more precise.
  2. Wearable/flexible devices and the healthcare system are two very hot areas. I think more popular literature should be cited. Like the impressive work published in related journals such as Nature Materials, Science Advance, Advanced Materials… etc.
  3. Keywords should be simplified.
  4. Figure 2 is not clear. Please prepare better pictures.
  5. Please double-check the chapter context of the manuscript. Part 2 is NFC, Part 4 is NFC sensor for biochemical and biophysical signals monitoring. But Part 3 is wearable materials.
  6. For flexible devices, weak electrical signal processing and display are very important, which is rarely mentioned in the manuscript. So I disagree with the author’s statement in part summary and perspectives, “Here, we provide a comprehensive overview of all the key components required for the next generation of wearable wireless sensing technology.

Author Response

Thank you very much for your time involved in reviewing the manuscript. Thank you for the many insightful comments and suggestions. We have made revisions to address all the comments. A list of our responses to each of the specific comments is listed below.

Reviewer 2 Report

This work by Sun et al. provides in-depth detail of the fabrication, near field communication and applications of wearable sensors. In particular, the author introduces the field by discussing the difference between RFID, NFC and Bluetooth. The authors address the challenges of designing wearable designers, including compliance, energy requirements and material selection. The review finalizes with different case studies of wearable devices for health care monitoring.

In general, the review is well structured and up to date references. Therefore, I would accept this article for publication after addressing the following minor clarifications.

  1. The authors need to expand the summary and perspective section. The current paragraph is too short and superficial. Nevertheless, the authors are likely to master state-of-the-art in the field and could provide valuable insight into the future of the field and what are the most exciting applications in medical monitoring.
  2. I would suggest providing a recap in the summary and perspective section, discussing the unique challenges and opportunities of the different types of RFID, NFC and Bluetooth.

Author Response

Thank you very much for your time involved in reviewing the manuscript. Thank you for the many insightful comments and suggestions. Those comments are all valuable and very helpful for revising and improving our paper. We have studied comments carefully and have made correction which we hope meet with approval.
